# Anti-Acetylcholinesterase Activities of Mono-Herbal Extracts and Exhibited Synergistic Effects of the Phytoconstituents: A Biochemical and Computational Study

**DOI:** 10.3390/molecules24224175

**Published:** 2019-11-18

**Authors:** Acharya Balkrishna, Subarna Pokhrel, Meenu Tomer, Sudeep Verma, Ajay Kumar, Pradeep Nain, Abhishek Gupta, Anurag Varshney

**Affiliations:** 1Drug Discovery and Development Division, Patanjali Research Institute, Patanjali Research Foundation Trust, Roorkee-Haridwar Road, Haridwar 249405, India; pyp@divyayoga.com (A.B.); meenu.tomer@prft.co.in (M.T.); sudeep.verma@prft.co.in (S.V.); ajay.kumar@prft.co.in (A.K.); pradeep.nain@prft.co.in (P.N.); abhishek.gupta@prft.co.in (A.G.); 2Department of Allied Sciences, University of Patanjali, Roorkee-Haridwar Road, Haridwar 249405, India

**Keywords:** Alzheimer’s disease, anti-acetylcholinesterase activity, herbal extracts, HPLC, synergistic effect, molecular docking

## Abstract

Alzheimer’s disease (AD), a neurodegenerative disease, is the most common form of dementia. Inhibition of acetylcholinesterase (AChE) is a common strategy for the treatment of AD. In this study, aqueous, hydro-methanolic, and methanolic extracts of five potent herbal extracts were tested for their in vitro anti-AChE activity. Among all, the *Tinospora cordifolia* (Giloy) methanolic fraction performed better with an IC_50_ of 202.64 µg/mL. Of the HPLC analyzed components of *T. cordifolia* (methanolic extract), palmatine and berberine performed better (IC_50_ 0.66 and 0.94 µg/mL, respectively) as compared to gallic acid and the tool compound “galantamine hydrobromide” (IC_50_ 7.89 and 1.45 µg/mL, respectively). Mode of inhibition of palmatine and berberine was non-competitive, while the mode was competitive for the tool compound. Combinations of individual alkaloids palmatine and berberine resulted in a synergistic effect for AChE inhibition. Therefore, the AChE inhibition by the methanolic extract of *T. cordifolia* was probably due to the synergism of the isoquinoline alkaloids. Upon molecular docking, it was observed that palmatine and berberine preferred the peripheral anionic site (PAS) of AChE, with π-interactions to PAS residue Trp286, indicating that it may hinder the substrate binding by partially blocking the entrance of the gorge of the active site or the product release.

## 1. Introduction

Alzheimer’s disease (AD) is a neurodegenerative disease and is the most common form of dementia in elderly people worldwide [1,2]. Impaired speech comprehension, poor coordination, and reduced executive functions are the characteristics of AD patients. The major histo-pathological signs of AD are the neurofibrillary tangles (NFTs) and senile plaques, consisting of proteinous aggregates of hyperphosphorylated tau protein and amyloid β (Aβ) of different sizes, respectively [3]. There is consistent cholinergic deficit due to the degeneration or atrophy of cholinergic neurons in the basal forebrain, along with the presence of NFTs and senile plaques [4]; therefore rebalancing cholinergic input is a way to increase memory and cognition in AD patients [5]. Acetylcholinesterase (AChE; EC 3.1.1.7) acts on the neurotransmitter acetylcholine (ACh), which mediates cholinergic transmission by the activation of ionotropic nicotinic and metabotropic muscarinic receptors. Therefore, the inhibitors of AChE can enhance cholinergic transmission by limiting ACh degradation. The cognitive impairment is due to the loss of ACh, which results from the hydrolytic action of AChE. Therefore, the emphasis has remained on anticholinergic drugs, which can inhibit the enzymes and up-regulate the level of ACh [6]. The modulation of AChE is currently the most promising therapeutic means for the development of cognitive enhancers [7].

Tacrine (Cognex™), the first FDA-approved medicationfor AD, was an AChE inhibitor. However it was withdrawn from the market because of its hepatotoxicity [8]. The most commonly prescribed AChE inhibitors are donepezil (Aricept™), rivastigmine (Exelon™), and galantamine (Reminyl™), which were approved in 1996, 2000, and 2001, respectively [9,10]. Galantamine and rivastigmine are plant derived alkaloids. Among these inhibitors, Donepezil is the only AChE inhibitor approved for the treatment of all stages of AD [8,9,10]. Moreover, ‘‘cholinergic hypothesis’’ has emerged as a widely accepted therapeutic means for improving cognitive functions in AD [11]. AChE inhibition has also been recognized as a therapeutic strategy for other types of disorders such as dementia, myasthenia gravis, glaucoma, and Parkinson’s disease in addition to AD [12].

In the present work, we have studied fifteen fractions of the five mono-herbal extracts (*Bacopa monnieri* (L.) Wettst.—Brahmi (leaves), *Withania somnifera* (L.) Dunal—Ashwagandha (roots), *Convolvulus pluricaulis* Choisy—Shankhpushpi (whole plant), *Celastrus paniculatus* (Willd.)—Malkagni (seeds), and *Tinospora cordifolia* (Wild.) Hook. f. & Thoms.—Giloy (stems) against AChE. “Brahmi” has been recognized in Ayurveda as a remedy for restoring memory and is believed to be beneficial for longevity. It has been used to retard the symptoms of ageing and in preventing dementia [13]. The Indian medicinal plant *Withania somnifera* (L.) Dunal; family Solanaceae), commonly known as Ashwagandha, is widely used as an Ayurvedic medicine to combat stress, arthritis, inflammations, conjunctivitis, and tuberculosis. Sitoindisides VII–X and withaferin-A are the active ingredients, which have shown significant anti-stress and anti-oxidant properties [14,15]. Phyto-reservoirs have already proven to be promising sources of AChE inhibitors [16,17], but there is still a need for newer potent AChE inhibitors with minimal side effects. Hence, in this work, we have studied comparatively the three different fractions (aqueous, hydro-methanolic, and methanolic) of five potent anti-acetylcholinesterase herbal extracts; that is why the outcome of this work could be useful for lead compound identification for cognitive function improvement in AD.

## 2. Results and Discussion

Nature has a high diversity of phyto-compounds that might be beneficial for the treatment of various human diseases. Therefore, this study was designed to evaluate the potential of five potent herbal extracts as inhibitors of acetylcholinesterase, a well-known Alzheimer’s disease target. Plant secondary metabolites have shown anti-cholinesterase activities, including alkaloids, flavonoids, and lignans. Alkaloids are the largest group of AChE inhibitors [18,19]. Earlier reports have shown that the stronger inhibitory activity of alkaloids is associated with their similarity to ACh, and many alkaloids have positively-charged nitrogen which can bind in the gorge of active sites of AChE [20]. Therapies based on inhibitors of AChE are supposed to reverse cholinergic deficits in AD [21]. Galantamine, an isoquinoline alkaloid family, is a reversible and competitive inhibitor of AChE. It increases the level of ACh in the synaptic cleft, thus improving cholinergic transmission and improving neuron to neuron communications [22]. It has a dual action of mechanism; inhibiting AChE and allosterically modulating nicotinic acetylcholine receptor (nAChR) activity [23]. Galantamine shows 53-fold higher selectivity for human AChE than for butyrylcholinesterase (BChE) [24]. Despite of many reports, there is still a need to explore new AChE inhibitors with lower toxicity and higher central nervous system (CNS) penetration. Ayurveda is the ancient Indian system of medicine which dates back to 2000 BC, in which various plants/parts effective for treating CNS disorders and aging are well documented [25]. In the present study, aqueous, hydro-methanolic, and methanolic extracts of five plant materials considered to be “nootropic” or brain boosting were prepared and evaluated for their anti-acetylcholinesterase effects using Ellman’s colorimetric method.

### 2.1. Determination of Acetylcholinesterase Inhibitory Activity (Screening)

The results are shown in Table 1 and Figure 1. At the concentration of 100 µg/mL, the aqueous, hydro-methanolic, and methanolic extracts of *W. somnifera* showed higher AChE inhibitory activities (with inhibition of 24.26%, 31.47%, and 30.03%, respectively). At the same concentration, the aqueous and methanolic extracts of *T. cordifolia* displayed 26.01% and 40.58% inhibition, respectively. The rest of the extracts showed AChE inhibitory activity below 20%. Galantamine hydrobromide was used as a standard AChE inhibitor in this study. Galantamine, a known acetylcholinesterase inhibitor [26], belongs to the isoquinoline alkaloid family. At the concentration of 10 µg/mL, galantamine showed 94.33% inhibition. *T. cordifolia* (methanolic fraction) performed better as compared to the rest of the fractions. There is a report by Mathew and Subramanian (2014) [27] for AChE inhibition by 20 medicinally important plants used for cognitive disorders. As per their report, methanolic fractions of *B. monnieri*, *W. somnifera*, *C. pluricaulis*, and *C. paniculatus* had 30.7%, 44.8%, 40.6%, and 23.13% inhibitory activities against AChE from electric eel.

### 2.2. IC_50_ Determination of the Extracts/Components and the Positive Control

The half maximal inhibitory concentration (IC_50_) values of the extracts, components, and the positive control have been shown in Table 1 and Figure 2A–D. Galantamine hydrobromide showed IC_50_ of 1.45 µg/mL (3.95 µM). IC_50_ of galantamine for AChE was reported to be 0.35 µM, whereas it was 4.53 mg/mL (against AChE from electric eel) as per the report of Kaufmann et al. (2016) [28]. *W. somnifera* (aqueous extract) had an IC_50_ of 540.98 µg/mL, whereas the IC_50_ of *T. cordifolia* (aqueous extract) was 930.06 µg/mL. Other results have been shown in Table 2 and Figure 2A–C. *W. somnifera* (hydro-methanolic extract) showed IC_50_ of 306.72 µg/mL, whereas the IC_50_ of *W. somnifera* (methanolic extract) was 203.79 µg/mL. *T. cordifolia* (methanolic extract) performed better among all the extracts, its IC_50_ was 202.64 µg/mL. *W. somnifera* (methanolic extract) has an IC_50_ of 4.44 mg/mL and 124 µg/mL, as shown by Pal et al. (2017) [29] and Mathew and Subramanian (2014) [27], respectively.

There are many reports on *W. somnifera* (methanolic extract) [24,26], and the IC_50_ of *T. cordifolia* (methanolic extract) was lowest in the study group, therefore we further analyzed it by HPLC. We determined IC_50_ of the components like palmatine, gallic acid, and berberine, and also, we analyzed vanillic acid and ferulic acid against AChE. Vanillic acid and ferulic acid displayed no inhibition up to the concentration of 50 µg/mL. Therefore we did not proceed further with those two acids. The IC_50_ of palmatine was 0.66 µg/mL (1.93 µM), whereas the IC_50_ of berberine was 0.94 µg/mL (2.80 µM); both the inhibitors showing stronger interaction as compared to galantamine hydrobromide (IC_50_ 1.45 µg/mL or 3.95 µM) against AChE. However, gallic acid showed higher IC_50_ (7.89 µg/mL, 46.39 µM). However, it has been noticed that the IC_50_ values of palmatine for inhibiting AChE determined by different researchers were quite different, including 3.80 µM and 5.21 mg/mL (AChE from electric eel), 0.46 µM (AChE from human) and 0.74, 2.20, 36.6 µM and 14.82 µg/mL (the origin of AChE is unknown) [30].

Berberine, a plant isoquinoline alkaloid, is commonly found in the roots, bark, and stems of several medicinal plants including *Berberis spp*., *Hydrastis canadensis*, and *Coptidis* rhizome [31]. Studies show that berberine exhibits beneficial effects in AD through inhibition on both AChE and butyrylcholinesterase (BChE) [32,33,34] with IC_50_ of 0.37 µM and 18.21 µM, respectively. These results suggest that berberine enhances cholinergic stimulation and can therefore be used to improve cognitive impairment in AD. It has been shown that palmatine can be used to treat Alzheimer’s disease (AD), mainly by inhibiting the activity of AChE, BChE and neuraminidase-1 (NA-1) [20].

### 2.3. HPLC Analysis of the Methanolic Extract of Tinospora cordifolia

After the screening and IC_50_ determination, the most potent extract, “*T. cordifolia* (methanolic extract)”, was subjected to HPLC analysis. HPLC analysis showed 0.134 mg/g gallic acid, 0.159 mg/g palmatine, 0.022 mg/g berberine, 0.494 mg/g vanillic acid, and 0.205 mg/g ferulic acid (Table 2). The chromatographs have been presented in Figure 3A,B (refer to Appendix A 1 for the details).

The quantification of the phyto-compounds was performed by an in-house developed and validated HPLC method (refer to Section 3.5 for details).

### 2.4. Inhibition Kinetics

The inhibition modes of the positive control and the components were analyzed by double reciprocal Lineweaver–Burk plots. Lineweaver–Burk plots of the tool compound, galantamine hydrobromide, were fitted to be competitive (Table 3, Figure 4A). It showed increased K_m_ (1.87 mM) and unaffected V_max_(360 U/mg), whereas the other two inhibitors, namely, palmatine and berberine, were non-competitive (Table 3, Figure 4B,C). Palmatine and berberine had unaffected K_m_ (0.38 and 0.36 mM, respectively) and reduced V_max_ (90.86 and 72.65 U/mg, respectively).

### 2.5. Evaluation of Synergistic Effects

Of the HPLC analyzed components of *T. cordifolia* (methanolic extract), only the palmatine and berberine displayed pronounced inhibitory effects against AChE compared to galantamine. Therefore, these two compounds were further evaluated. The effects of palmatine and berberine, and their combination on AChE (electric eel) were evaluated by Ellman’s assay. Various doses of palmatine or berberine (0.05–50 μg/mL) and either of them at fixed IC_20_ was used to evaluate their AChE inhibitory effects. The combination of palmatine (varying) and berberine (IC_20_) more synergistically inhibited AChE than the combination of berberine (varying) and palmatine (IC_20_) (Figure 5). Both the combinations synergistically inhibited AChE at fa 0.5.

### 2.6. Molecular Docking

Molecular docking was performed using ADV [35] to access the binding position of the palmatine and berberine onto the AChE. First of all, re-docking of galantamine was performed to validate the method by considering 10 crystallographic water molecules. The RMSD of the docked and the crystallized position of galantamine was 0.188 Å, showing the perfect docking protocol. After validating the methodology, palmatine and berberine were docked into the same initial location as that of galantamine. The preference of palmatine and berberine was different as compared to that of galantamine (binds at the bottom of the deep gorge of the active site); they were bound to the peripheral anionic site (PAS) in the AChE molecule (Figure 6A). The PAS consists of 5 residues (Tyr72, Asp74, Tyr124, Trp286, and Tyr341) clustered around the entrance to the active site gorge. There is an important role of PAS as it binds substrate transiently at the first step of the catalytic pathway, enhancing catalytic efficiency by trapping substrate on its way to the active site [36]. Both the palmatine and berberine showed π-interaction to Trp 286 (Figure 6B,C, respectively) in PAS of the AChE molecule.

The non-competitive nature of inhibition of palmatine and berberine (Figure 4B,C, respectively) against AChE points towards a different binding site in contrast to galantamine (competitive inhibitor) (Figure 4A). This is well supported by the docked poses (Figure 6A). The stronger inhibition data (experimental) of palmatine and berberine (Figure 2D), and the binding of palmatine and berberine at PAS with π-interactions to PAS residue Trp286 (Figure 6B,C, respectively) indicate that these compounds may hinder the substrate binding by partially blocking the entrance of the gorge of the active site or the product release processes.

## 3. Materials and Methods

### 3.1. Plant Materials, Chemicals and Reagents

Plant materials were sourced from Divya Pharmacy, Haridwar, India, and were independently certified by Council of Scientific and Industrial Research—National Institute of Science Communication and Information Resources (CSIR—NISCAIR), New Delhi, India. Identification voucher numbers of *B. Monnieri, W. somnifera, C. Pluricaulis, C. Paniculatus* and *T. cordifolia* were NISCAIR/RHMD/Consult/2019/3453-54-32, 3453-54-15, 3453-54-167, 3453-54-120, and 3453-54-63, respectively. Acetylthiocholine chloride (ATCC), acetylcholinesterase from *Electrophorus electricus* (electric eel) (type VI-S, lyophilized powder, CAS Number: 9000-81-1), 5, 5-dithiobis [2-nitrobenzoic acid] (DTNB), and galantamine hydrobromide were purchased from Sigma (India). All other reagents were of analytical grade and purchased locally.

### 3.2. Preparation of Extracts

20 g each of powdered *Bacopa monnieri* (leaves), *Withania somnifera* (roots), *Convolvulus pluricaulis* (whole plant), *Celastrus paniculatus* (seeds), and *Tinospora cordifolia* (stems) was mixed with 200 mL of distilled water (for aqueous extracts) (200 mL × 2 wash), 200 mL of methanol (for methanolic extracts) (200 mL × 2 wash), and 200 mL of hydro-methanol (for hydro-methanolic extract) (200 mL × 2 wash with 1:1 ratio), respectively, and each mixture was refluxed at 60 °C on a heating mantle with a condenser for 2 h. It was then cooled, filtered, and the filtrate was collected. Then, the whole filtrate was concentrated on a rota-evaporator up to complete dryness.

### 3.3. Determination of Acetylcholinesterase Inhibitory Activity (Screening)

AChE activity was measured by Ellman’s method [37], with slight modifications in the protocol. In the procedure, the enzyme hydrolyses the substrate acetylthiocholine, resulting in the product “thiocholine” which reacts with Ellman’s reagent (5,5′-dithio-bis-(2-nitrobenzoic acid), DTNB) to produce 2-nitrobenzoate-5-mercaptothiocholine and 5-thio-2-nitrobenzoate (TNB); the latter can be detected at 412 nm. Phosphate buffer (pH 8) of 100 mM was used as a buffer throughout the experiment. The AChE used in the assay was from electric eel (type VI-S lyophilized powder, 8.52 mg solid). The enzyme stock solution was prepared at a concentration of 20 mg/mL in 0.1% BSA in phosphate buffer and was kept at −20 °C until assayed. For a working enzyme solution, the enzyme was diluted 1:100,000 times in 0.1% BSA buffer. ATCC was dissolved in the buffer. DTNB and SDS mixture was prepared at a concentration of 15 mM and 10% respectively in 50% ethanol. The conditions for reaction time (10 to 40 min), enzyme concentration (0.8 to 6.4 ng), ATCC concentration (0.25 to 1 mM) and DTNB concentration (0.3 to 1.2 mM) were optimized in a 200 µL reaction volume. 20 µL of the enzyme (4 ng) and 10 µL of the substrate (0.5 mM final concentration) were used in the assay. Reaction was performed at 37 °C by taking an appropriate volume of buffer, 10 µL of the extract (2 mg/mL in water, hydro-methanol, and methanol for aqueous, hydro-methanolic, and methanolic extracts)/positive control-galantamine hydrobromide (0.2 mg/mL in water). The extracts/positive control were allowed to bind to AChE for 30 min, and then, 10 µL of ATCC (10 mM) was added to initiate the reaction for a period of 20 min. The reaction was stopped and colour was developed by adding 10 µL of DTNB/SDS mixture, and incubated for further 10 min. After mixing, the absorbance was measured at 412 nm using EnVision multimode plate reader (PerkinElmer, Waltham, MA, USA). All the reactions were performed in triplicates. The activity of the negative control was also examined with and without an inhibitor. The inhibitory activity (I) was calculated according to the following formula:Inhibitory activity (I%) = [(*A* − *a*) − (*B* − *b*)]/(*A* − *a*) × 100(1)
where, *A* is the activity without inhibitor; *a* is the negative control without inhibitor; *B* is the activity with inhibitor; and *b* is the negative control with inhibitor.

### 3.4. IC_50_ Determination of the Extracts/Components and the Positive Control

The extracts were serially diluted in the buffer to prepare 10,000 to 10 µg/mL final concentrations (100 to 0.01 µg/mL for positive control) in a reaction system of 200 µL. 10 µL of the extracts/positive control were pre-incubated with the enzyme (4 ng) for 30 min at 37 °C. After the incubation period, 10 µL ATCC solutions (10 mM) were added and the mixture was incubated again for 20 min at 37 °C. The reaction was terminated and color was developed by adding 10 µL of DTNB/SDS solution, and the mixture was further continued to incubate for 10 min. The hydrolysis of ATCC was monitored at 412 nm using an EnVision multimode plate reader (PerkinElmer). All the experiments were performed in triplicates. The activity of the negative control was also examined with and without an inhibitor. The inhibitory activity (I) was calculated according to the following formula given below and IC_50_ was determined using GraphPad Prism 7.
Inhibitory activity (I%) = [(*A* − *a*) − (*B* − *b*)]/(*A* − *a*) × 100(2)
where, *A* is the activity without inhibitor; *a* is the negative control without inhibitor; *B* is the activity with inhibitor; and *b* is the negative control with inhibitor.

### 3.5. HPLC Analysis of the Methanolic Extract of Tinospora cordifolia

Half gram (0.5 g) of the *T. cordifolia* extract was dissolved in 10 mL methanol and sonicated for 30 min, centrifuged (8000 rpm, 5 min) and filtered using 0.45 µm nylon filter. The filtered solution was used for further analysis. Standards were purchased from Natural Remedies Pvt. Ltd. and Sigma Aldrich (Bangalore, India).

The analysis was performed by an in-house developed and validated method on a Waters HPLC system equipped with a binary pump (1525), PDAD (2998) and auto-sampler (2707). The elution was carried out at a flow rate of 1.0 mL/min using gradient elution of mobile phase A (0.140 g of KH₂PO₄ dissolved in 1000 mL of water, pH 2.5 with orthophosphoric acid) and mobile phase B (acetonitrile). This experiment was performed on a Shodex C18-4E (4.6mm ID × 250 mm L) column and column temperature was kept at 35 °C during the analysis. Injection volume was 10 µL and the chromatograph was recorded at 346 and 270 nm wavelength (refer to Appendix A for the details).

### 3.6. Inhibition Kinetics

Inhibition kinetics study of the HPLC analyzed most potent components (of *T. cordifolia* methanolic fraction) was performed to draw Lineweaver–Burk plots. The positive control- galantamine hydrobromide was prepared in buffer and the working solutions of the phyto-compounds (palmatine and berberine) were prepared in methanol, so as to give a final concentration of 1 and 5 µg/mL in a reaction system of 200 µL. The substrate (ATCC) concentration was varied from 0.1 to 1.6 mM final concentrations, and the rest of the conditions and methodology were same as stated above in the “screening” section. The reaction velocities were determined using a molar extinction coefficient of TNB (14,150 M^−1^ cm^−1^) at 412 nm [38]. K_m_ and V_max_ were deduced by Michaelis–Menten plots using GraphPad Prism 7.

### 3.7. Evaluation of Synergistic Effects

It was clear by the screening of the phyto-extracts and the components that the distinctive inhibitory activity of the extracts—like *T. cordifolia* (methanolic fraction)—must be based on synergism of individual components present in the extracts. Therefore, synergism studies were carried out with its component alkaloids “berberine” and “palmatine”. First of all, the IC_20_ values for berberine and palmatine alone were determined. For berberine, a serial dilution (0.05 to 50 µg/mL final concentrations for a 200 µL reaction system) was prepared and palmatine at a fixed concentration (IC_20_) was added. After this, the AChE inhibition assay was carried out as described above. For palmatine, a serial dilution (0.05 to 50 µg/mL final concentrations in a 200 µL reaction system) was prepared and berberine at a fixed concentration (IC_20_) was added, and the AChE inhibitory activity was determined as stated above. Drug interaction was classified as synergistic [combination index (CI) < 1], additive (CI = 1) or antagonistic (CI > 1) based on Chou–Talalay equation (1984) [39], as solved by CompuSyn [40].

### 3.8. Molecular Docking

Human AChE structure in complex with galantamine (PDB ID: 4EY6, 2.4 Å resolution) was obtained from the Protein Data Bank (http://www.rcsb.org). The AChE chain A was selected for docking study by editing on PyMOL [41]. Crystallographic water molecules within 4 Å around the crystallized position of galantamine (10 water molecules) were not removed. The galantamine structure was retrieved from the crystal structure of AChE. The 3D structures of palmatine and berberine were downloaded from PubChem (https://pubchem.ncbi.nlm.nih.gov). The SDF files of the ligands downloaded from the database were then converted into PDB files using OpenBabel 2.4.1 [42]. The binding cavity was determined based on the binding location of galantamine in the native pose. The dock prep module of the UCSF Chimera-1.13.1 [43] was used for all the structure preparations.

Molecular docking was performed using AutoDock Vina-1.1.2 (ADV-1.1.2) [36]. For intermediary steps, such as PDBQT files for protein and ligands preparation and grid box creation, were performed using a Graphical User Interface program AutoDock Tools-1.5.6 (ADT-1.5.6) [44]. ADT was used to assign polar hydrogens and Gasteiger charges. The “choose ligand” option was used to set map file types. AutoDock was used to save the prepared files in PDBQT format. Grid maps were prepared using a grid box size of 30 × 30 × 30 xyz points and the protein center (x = −10.492, y = −43.526, z = 29.454). To obtain the maximum number of poses, we set the num_modes to 20 and the energy range to 9, and exhaustiveness was set to 8. The pose with lowest energy of binding was extracted and aligned with the receptor for further analysis by Discovery Studio 2017 R2 Client [45]. Docking was validated by re-docking the galantamine into AChE, rendering the side chains of the residues within 4 Å of the galantamine (Asp74, Trp86, Gly120, Gly121, Gly122, Tyr124, Ser125, Glu202, Ser203, Phe295, Phe297, Tyr337, Phe338, and His447) in 4EY6 as flexible. Docking simulations were initiated with random seed.

### 3.9. Statistical Analyses and Graphics

IC_50_ was determined using 4PL (4 parameter logistic) non-linear regression analysis by GraphPad Prism 7. Lineweaver–Burk plots were drawn using linear regression analysis by GraphPad Prism 7. All the graphics related to IC_50_ and enzyme kinetics were generated using GraphPad Prism 7.

## 4. Conclusions

Phyto-compounds are useful in the treatment of Alzheimer’s disease which indicates that nature is a valuable source of new bioactive agents. We evaluated various extracts for their potential AChE inhibitory activities. Upon comparing their IC_50_ values, it was clear that the methanolic extracts of *T. cordifolia* and *W. somnifera* were much more active than the other extracts. Of the HPLC analyzed components of *T. cordifolia* (methanolic extract), palmatine and berberine were more active against AChE. This study points out that the AChE inhibitory effect of the alkaloids palmatine and berberine is clearly synergistic in nature. Molecular docking revealed that the palmatine and berberine had a binding preference to PAS, which could lead to partial substrate blocking or hindrance in the product release process. Despite having higher IC_50_ values compared to the tool compound, these mono-herbal extracts (which may contain thousands of different phyto-compounds) were showing potency against AChE, which should be further evaluated and could be used as cost-effective alternatives for the treatment of AD.

## Figures and Tables

**Figure 1 molecules-24-04175-f001:**
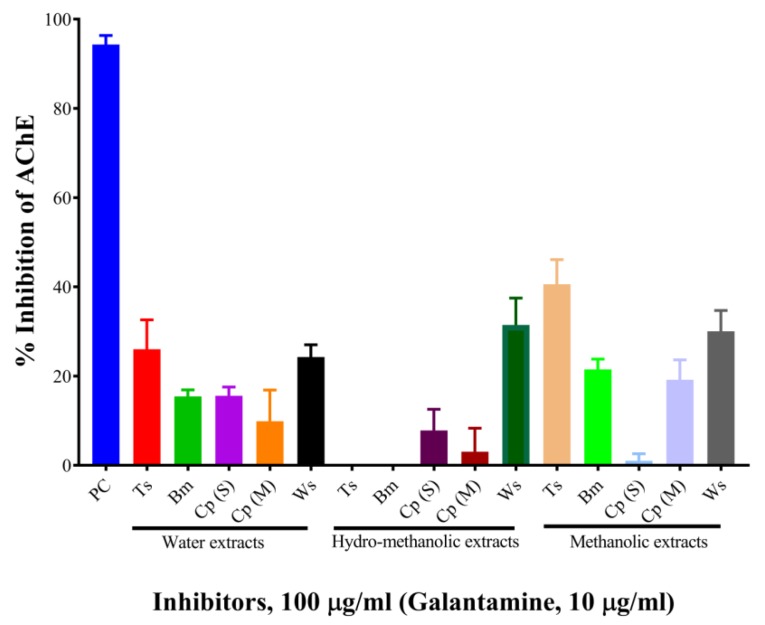
Screening of mono-herbal extracts against AChE. Experiments were performed in triplicates. PC, positive control (galantamine hydrobromide); Ts, *Tinospora cordifolia* (Giloy); Bm, *Bacopa monnieri* (Brahmi); Cp (S), *Convolvulus pluricaulis* (Shankhpushpi); Cp (M), *Celastrus paniculatus* (Malkagni), Ws, *Withania somnifera* (Ashwagandha).

**Figure 2 molecules-24-04175-f002:**
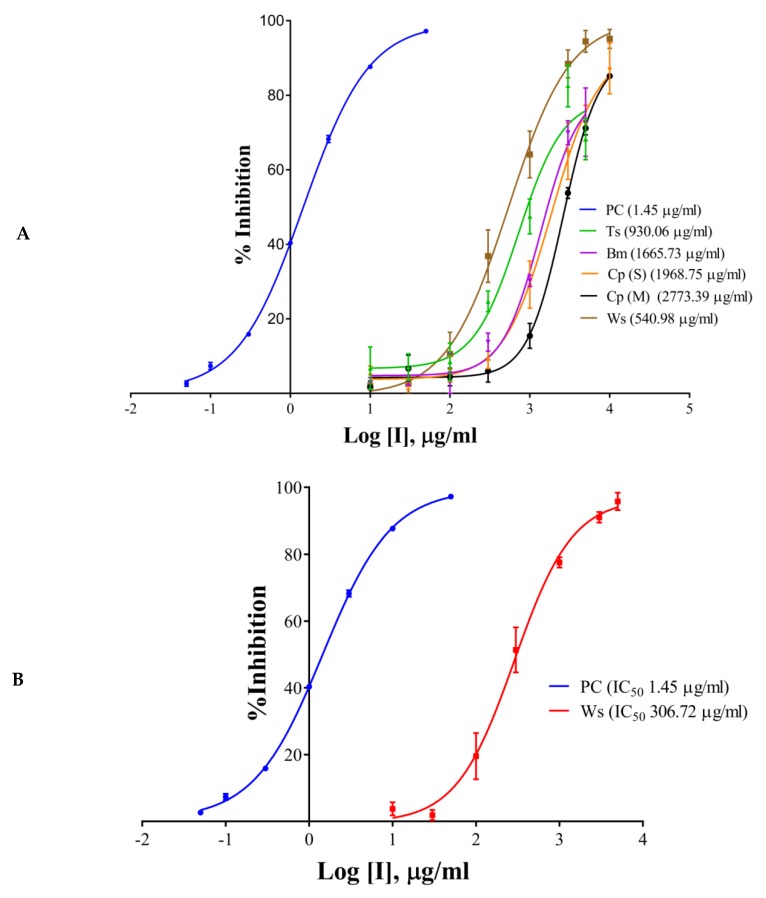
IC_50_ curves of (**A**) aqueous extracts, (**B**) hydro-methanolic extract, (**C**) methanolic extracts, of different plant parts, and (**D**) HPLC analyzed components of *T. cordifolia* (methanolic fraction) with positive control (PC) against AChE. IC_50_ values are shown in brackets. Experiments were performed in triplicates. PC (positive control), galantamine hydrobromide; Ts, *Tinospora cordifolia* (Giloy); Bm, *Bacopa monnieri* (Brahmi); Cp (S), *Convolvulus pluricaulis* (Shankhpushpi); Cp (M), *Celastrus paniculatus* (Malkagni), Ws, *Withania somnifera* (Ashwagandha).

**Figure 3 molecules-24-04175-f003:**
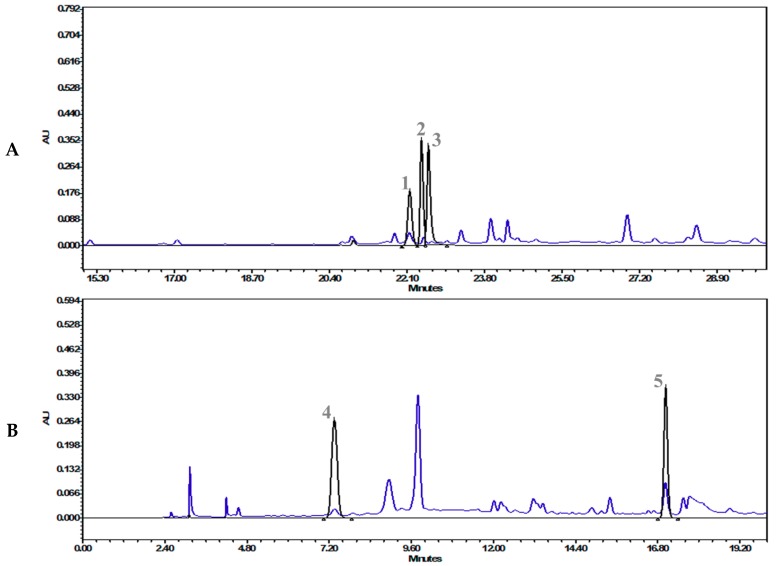
Overlap chromatographs of (**A**) the standard mix of ferulic acid (1, 22.16 min), palmatine (2, 22.42 min) berberine (3, 22.57 min), and *T. cordifolia* (Giloy)–methanolic extract at 346 nm, and (**B**) the standard mix of gallic acid (4, 7.35 min), vanillic acid (5, 17.03 min), and *T. cordifolia* (Giloy)–methanolic extract at 270 nm.

**Figure 4 molecules-24-04175-f004:**
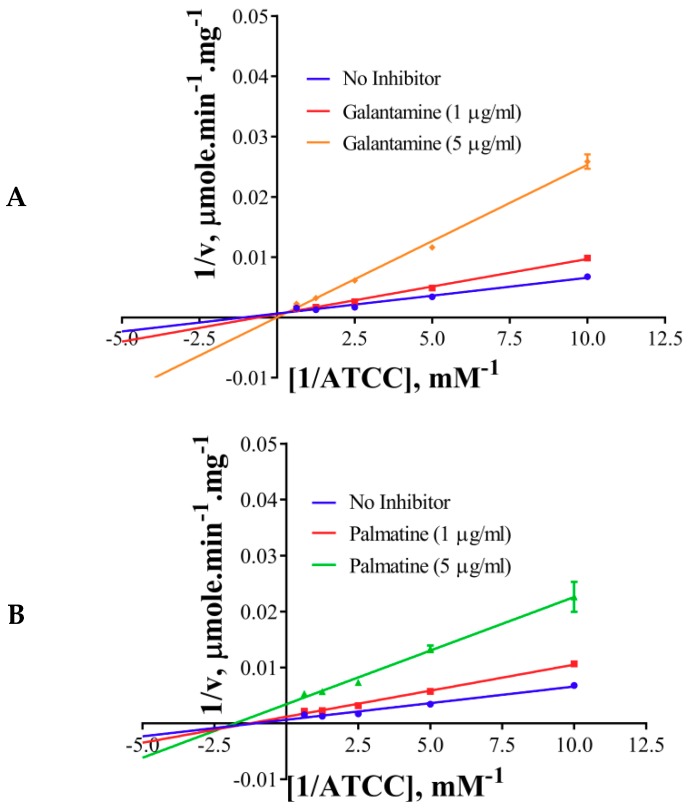
Lineweaver–Burkplots of AChE (**A**) in the absence or presence of galantamine hydrobromide, (**B**) in the absence or presence of palmatine, and (**C**) in the absence or presence of berberine. Experiments were performed in triplicates.

**Figure 5 molecules-24-04175-f005:**
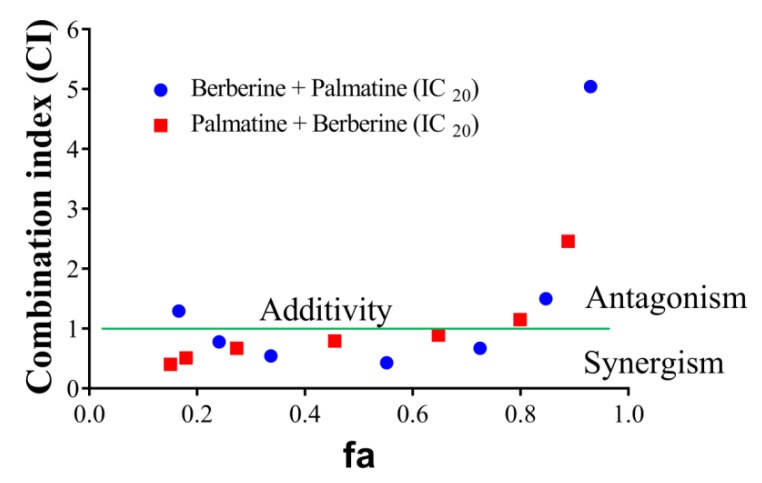
CI-fa plot of palmatine and berberine. (●) varying berberine concentration from 0.05 to 50 µg/mL and fixed palmatine concentration (IC_20_), (●) varying palmatine concentration from 0.05 to 50 µg/mL and fixed berberine concentration (IC_20_).

**Figure 6 molecules-24-04175-f006:**
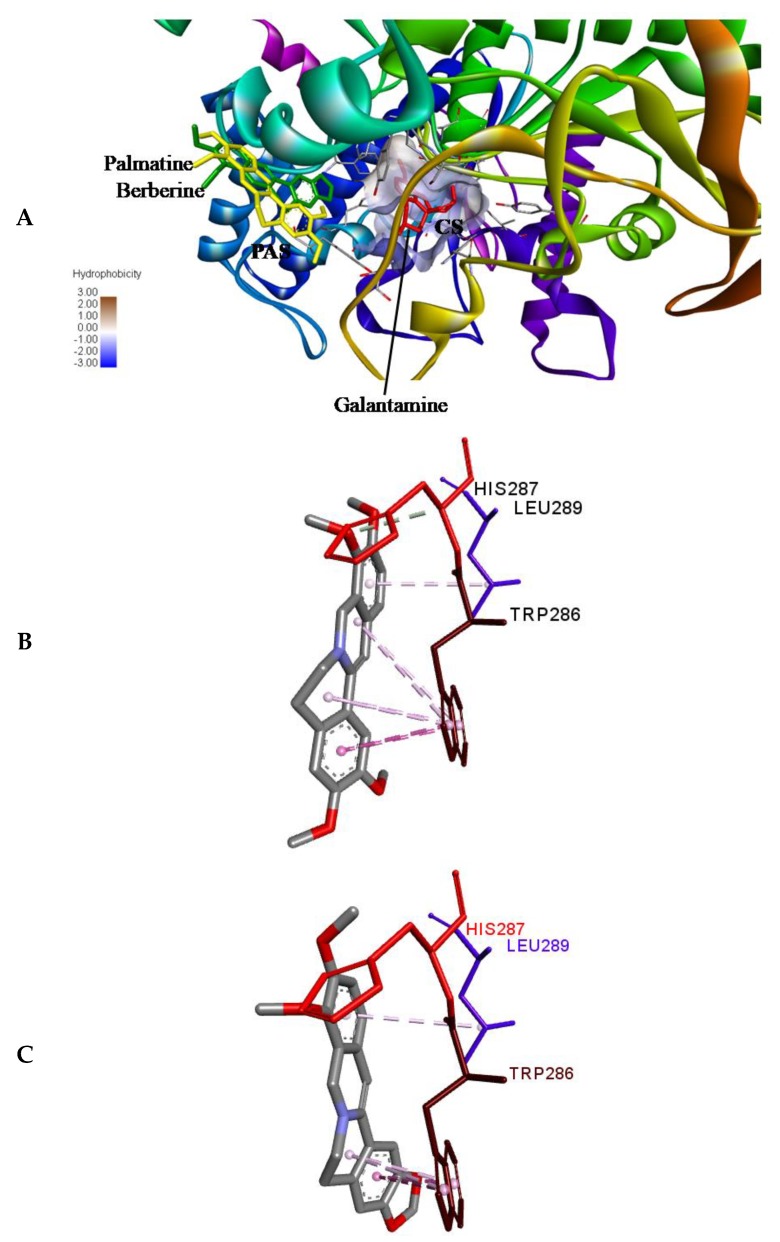
(**A**) Binding pose of palmatine, berberine, and galantamine (re-docked) onto AChE (PDB ID: 4EY6). Palmatine and berberine prefer the peripheral anionic site (PAS). CS, catalytic site. (**B**) Non-bonded interactions of (**B**) palmatine and (**C**) berberine in PAS; π-interaction to PAS residue Trp286 in AChE.

**Table 1 molecules-24-04175-t001:** Screening and IC_50_ determination of herbal extracts against AChE.

Inhibitors	% Inhibition at 100 µg/mL			IC_50_, µg/mL		
	A	HM	M	A	HM	M
*T. cordifolia* (Giloy)	26.01	0.00	40.58	930.06	ND	202.64
*B. monnieri* (Brahmi)	15.42	0.00	21.49	1665.73	ND	ND
*C. pluricaulis* (Shankhpushpi)	15.57	7.83	1.00	1968.75	ND	ND
*C. paniculatus* (Malkagni)	9.88	3.04	19.17	2773.39	ND	ND
*W. somnifera* (Ashwagandha)	24.26	31.47	30.03	540.98	306.72	203.79
Galantamine hydrobromide (at 10 µg/mL)	94.33	-	-	1.45	ND	ND

A = Aqueous, HM = Hydro-methanolic, M = Methanolic.

**Table 2 molecules-24-04175-t002:** HPLC analysis of *Tinospora cordifolia* (methanolic extract).

S.N.	Compound	Quantity, mg/g
1	Gallic acid	0.134
2	Palmatine	0.159
3	Berberine	0.022
4	Vanillic acid	0.494
5	Ferulic acid	0.205

**Table 3 molecules-24-04175-t003:** Kinetic parameters.

Inhibitor	K_m_, mM	V_max_, U/mg	Type of Inhibition
No inhibitor	0.33	340.8	NA
Galantamine	1.87	360	Competitive
Palmatine	0.38	90.86	Non-competitive
Berberine	0.36	72.65	Non-competitive

NA = Not applicable.

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
