# Peer review of "Anti-Acetylcholinesterase Activities of Mono-Herbal Extracts and Exhibited Synergistic Effects of the Phytoconstituents: A Biochemical and Computational Study"

_molecules, 2019, doi:10.3390/molecules24224175_

Round 1

Reviewer 1 Report

In the present manuscript, aqueous, hydro-methanolic and methanolic extracts of five potent herbal extracts (Bacopa monnieri – Brahmi (leaves), Withania somnifera – Ashwagandha (roots), Convolvulus pluricaulis – Shankhpushpi (whole plant), Celastrus paniculatus – Malkagni (seeds) and Tinospora cordifolia - Giloy (stems)) were tested for their in-vitro anti-AChE activity. Moreover, the major compounds were identified with liquid chromatography and possible interactions between the detected were evaluated with standard compounds, while molecular docking was also performed.

Overall, the manuscript is well written and presents significant results.

In conclusion, the methanolic extracts were not more effective than the other extracts for all the tested species but only in the case of Tinospora cordifolia and Withania somnifera. For example, aqueous extracts of Convolvulus pluricaulis were more effective than the other tested extracts. Moreover, considering that positive control exhibited considerably low IC50 values it is not safe to conclude that these extracts could be used as cost effective alternative for the treatment of Alzheimer’s disease  (see lines 406-408).

Author Response

Response to the comments on Manuscript “molecules-613491”

Reviewer 1

Comments and Suggestions for Authors

In the present manuscript, aqueous, hydro-methanolic and methanolic extracts of five potent herbal extracts (Bacopa monnieri – Brahmi (leaves), Withania somnifera – Ashwagandha (roots), Convolvulus pluricaulis – Shankhpushpi (whole plant), Celastrus paniculatus – Malkagni (seeds) and Tinospora cordifolia - Giloy (stems)) were tested for their in-vitro anti-AChE activity. Moreover, the major compounds were identified with liquid chromatography and possible interactions between the detected were evaluated with standard compounds, while molecular docking was also performed.

Overall, the manuscript is well written and presents significant results.

Point 1: In conclusion, the methanolic extracts were not more effective than the other extracts for all the tested species but only in the case of Tinospora cordifolia and Withania somnifera. For example, aqueous extracts of Convolvulus pluricaulis were more effective than the other tested extracts.

Response 1:

We agree that it is not appropriate to write “methanolic extracts were superior over others in inhibiting AChE”, rather the methanolic extracts of T. cordifolia and W. somnifera were more potent against AChE as compared to other extracts.

Changes in the manuscript (Line416): Upon comparing their IC50 values, it was clear that the methanolic extracts of T. cordifolia and W. somnifera were much more active than the other extracts.

Point 2: Moreover, considering that positive control exhibited considerably low IC50 values it is not safe to conclude that these extracts could be used as cost effective alternative for the treatment of Alzheimer’s disease  (see lines 406-408).

Response 2:

The tested mono-herbal extracts (though it may contain thousands of different phyto-compounds) have shown their potency against AChE. Obviously, galantamine have lower IC50 value compared to the values shown by these extracts. These extracts costs low price compared to the pure compound “galantamine”. Therefore, the extracts could be used as cost effective alternatives for the treatment of AD.

Changes in the manuscript (Lines421-424): The following sentence was added “Despite of having higher IC50 values compared to the tool compound, these mono-herbal extracts (may contain thousands of different phyto-compounds) were showing potency against AChE which should be further evaluated and could be used as cost effective alternatives for the treatment of AD”.

Reviewer 2 Report

Dear Editor,

According to my opinion it is not an appropriate contribution for Molecules.

I think it is much more suitable for journals, which are publishing compound isolation from plants.

Moreover, based on my knowledge, it is also presenting nothing new regarding cholinesterases and inhibition of those three selected compounds - galantamine, palmatine and berberine. The fact, that these compounds inhibit ChE is well known for many years. Also their mode of action is known. So, authors just repeated work made by others.

Moreover, conclusion, that ChE inhibition effect of these compounds is connected with AD is very limited. Based on ChE assay, authors could not conclude this. It is right only in case of galantamine - but this fact is known already many years. Du to this fact galantamine is considered to be used for this purpose.

BUT, rest two compounds - palmatine and berberine - have quaternary nitrogen in their molecules. So that, the main pharmacological postulate - that quaternary compounds do not cross Blood-brain barrier - is against the authors conclusion. These compounds will not pass BBB, or pass in very limited concentration, which will be not enough for treatment of AD.

Based on the above mentioned facts, I do not think, this article could be published in Molecules. 

Author Response

Response to the comments on Manuscript “molecules-613491”

Reviewer 2

Comments and Suggestions for Authors

Dear Editor,

According to my opinion it is not an appropriate contribution for Molecules. I think it is much more suitable for journals, which are publishing compound isolation from plants.

Moreover, based on my knowledge, it is also presenting nothing new regarding cholinesterases and inhibition of those three selected compounds - galantamine, palmatine and berberine. The fact, that these compounds inhibit ChE is well known for many years. Also their mode of action is known. So, authors just repeated work made by others.

Moreover, conclusion, that ChE inhibition effect of these compounds is connected with AD is very limited. Based on ChE assay, authors could not conclude this. It is right only in case of galantamine - but this fact is known already many years. Du to this fact galantamine is considered to be used for this purpose.

BUT, rest two compounds - palmatine and berberine - have quaternary nitrogen in their molecules. So that, the main pharmacological postulate - that quaternary compounds do not cross Blood-brain barrier - is against the authors conclusion. These compounds will not pass BBB, or pass in very limited concentration, which will be not enough for treatment of AD.

Based on the above mentioned facts, I do not think, this article could be published in Molecules. 

Response:

Phyto-compounds screening against disease target is an active area of research in drug discovery, and there are huge numbers of reports in this area, some of which have been reviewed in our manuscript also. We have followed the same to explore and compare five potent anti-AChE medicinal plants. Aqueous, hydro-methanolic and methanolic fractions of each medicinal plant were prepared and tested against AChE. This study provides more or less a complete biochemical data with molecular docking study. We have screened all the fifteen fractions against AChE, determined IC50 of the more potent ones, performed inhibition kinetics of the most potent ones and performed synergistic analysis as a part of wet-lab experimentations as a part of single project. Also, we have performed molecular docking analysis of palmatine and berberine against AChE.

This study opens a door for further study in the field, and to explore therapeutic potential of the extracts/phyto-compounds in treating CNS aetiologies. There are not many studies regarding the movement of palmatine and berberine across the blood brain barrier. Wang and co-workers (2005) administered Coptidis rhizoma extract intravenously at a dose of 10.2 mg/kg containing 3 mg/kg berberine to rats, the results showed that berberine could penetrate through the blood–brain barrier to reach the hippocampus, quickly distribute to the hippocampus, and slowly eliminate from the hippocampus, which suggests that berberine might directly act on certain regions of the hippocampus to provide a neuroprotective effect. Berberine in the hippocampus peaked at 3.67 h with a concentration of 272 ng/g. This study points towards the movement of berberine across the blood brain barrier in optimal quantity, and indicates possible therapeutic potential of berberine.

Reference:

Xueli Wang, Rufeng Wang, Dongming Xing, Hui Su, Chao Ma, Yi Ding, Lijun Du. Kinetic difference of berberine between hippocampus and plasma in rat after intravenous administration of Coptidis rhizoma extract. Life Sciences 77 (2005) 3058–3067

Changes in the manuscript (Lines421-424): The following sentence was added “Despite of having higher IC50 values compared to the tool compound, these mono-herbal extracts (may contain thousands of different phyto-compounds) were showing potency against AChE which should be further evaluated and could be used as cost effective alternatives for the treatment of AD”.

Reviewer 3 Report

I suggest the following correction in lines 45 and 46: ’the emphasis has remained on anticholinergic drugs’ should be written instead of ’has remained emphasis on anticholinergic drugs’.

In line 62 ’believed for longevity’ is not correct. I suggest the following modification: ’believed benefical for longevity’.

In line 75 the word ’design’ should be spelt ’designed’ correctly.

In the Figures 4A, B and C, ‘[1/ATCC], mM-1’ should be written on the horizontal axis instead of ‘[1/ATC], mM-1’.

The results obtained at 3.2 mM substrate concentration in Figures 4B and C should not be plotted because this concentration is too high for this representation, as it is not the case in Figure 4A. (In the Lineweaver-Burk representation, points near the substrate saturation cannot be taken into account in the line fitting.)
In lines 349 and 350 it can be read that ’The substrate (ATCC) concentration was varied from 0.05 to 3.2 mM final concentrations’, but the lowest concentration indicated in Figures 4 is 0.1 mM. Either the data at 0.05 mM in Figure 4 should be included or this section should be rewritten: ’The substrate (ATCC) concentration was varied from 0.1 to 1.6 mM final concentrations’.

In line 337 the abbreviation ’gm’ stands for gram which is grammatically correct, but earlier ’g’ was used for gram. I suggest that ’g’ should be used consequently in the whole article.

Author Response

Response to the comments on Manuscript “molecules-613491”

Reviewer 3

Comments and Suggestions for Authors

Point 1: I suggest the following correction in lines 45 and 46: ’the emphasis has remained on anticholinergic drugs’ should be written instead of ’has remained emphasis on anticholinergic drugs’.

Response 1: Corrected as advised

Changes in the manuscript (Line 45,46): ..... the loss of ACh, which results from the hydrolytic action of AChE. Therefore, the emphasis has remained on anticholinergic drugs... ……

Point 2: In line 62 ’believed for longevity’ is not correct. I suggest the following modification: ’believed beneficial for longevity’.

Response 2: Corrected as suggested

Changes in the manuscript (Line 64): .....believed beneficial for longevity……

Point 3: In line 75 the word ’design’ should be spelt ’designed’ correctly.

Response 3: Corrected as suggested

Changes in the manuscript (Line 77): …designed…

Point 4: In the Figures 4A, B and C, ‘[1/ATCC], mM-1’ should be written on the horizontal axis instead of ‘[1/ATC], mM-1’.

Response 4: Corrected as suggested

Changes in the manuscript: [1/ATCC], mM-1 written instead of [1/ATC], mM-1 in Figures 4A, B and C

Point 5: The results obtained at 3.2 mM substrate concentration in Figures 4B and C should not be plotted because this concentration is too high for this representation, as it is not the case in Figure 4A. (In the Lineweaver-Burk representation, points near the substrate saturation cannot be taken into account in the line fitting.)

Response 5: Figures were redesigned.

Changes in the manuscript: Figures 4B and C were redesigned and inserted into the manuscript by removing 3.2 mM substrate concentration data point.

Point 6: In lines 349 and 350 it can be read that ’The substrate (ATCC) concentration was varied from 0.05 to 3.2 mM final concentrations’, but the lowest concentration indicated in Figures 4 is 0.1 mM. Either the data at 0.05 mM in Figure 4 should be included or this section should be rewritten: ’The substrate (ATCC) concentration was varied from 0.1 to 1.6 mM final concentrations’.

Response 6:  Corrected as suggested

Changes in the manuscript (Lines 364): ….. concentration was varied from 0.1  to 1.6  mM final concentrations….

Point 7: In line 337 the abbreviation ’gm’ stands for gram which is grammatically correct, but earlier ’g’ was used for gram. I suggest that ’g’ should be used consequently in the whole article.

Response 7: Corrected as suggested

Changes in the manuscript (Line351): … elution of mobile phase A (0.140 g of….

Reviewer 4 Report

The manscript entitled “Anti-acetylcholinesterase activities of mono-herbal extracts and exhibited synergistic effects of phytoconstituents: a biochemical and computational study” is interesting and details significant results. However, the authors should rewrite the introduction because plagiarism was detected in that part and they should also clarify the plants identification (taxonomist and voucher number) and full scientific name.

They should also explain the methodology used in the quantification (Table 2).

Although the authors used a positive control they almost ignore it during the discussion. They should explain why the extract “can be an alternative for the treatment” having higher v IC50 values.

Author Response

Response to the comments on Manuscript “molecules-613491”

Reviewer 4

Comments and Suggestions for Authors

The manuscript entitled “Anti-acetylcholinesterase activities of mono-herbal extracts and exhibited synergistic effects of phytoconstituents: a biochemical and computational study” is interesting and details significant results.

Point 1: However, the authors should rewrite the introduction because plagiarism was detected in that part.  

Response 1:  The manuscript was rewritten after the detection of plagiarism.

Point 2: and they should also clarify the plants identification (taxonomist and voucher number) and full scientific name.

Response 2: Plant identification was done properly, and voucher number has been written.

Changes in the manuscript (Lines60-62):.... of the five mono-herbal extracts (Bacopa monnieri (L.) Wettst. – Brahmi (leaves), Withania somnifera (L.) Dunal – Ashwagandha (roots), Convolvulus pluricaulis Choisy – Shankhpushpi (whole plant), Celastrus paniculatus (Willd.) – Malkagni (seeds) and Tinospora cordifolia (Wild.) Hook. f. & Thoms. - Giloy (stems) against...

Changes in the manuscript (Lines275-280):.... Plant materials were sourced from Divya Pharmacy, Haridwar, India, and were independently certified by Council of Scientific and Industrial Research – National Institute of Science Communication and Information Resources (CSIR – NISCAIR), New Delhi, India.

Point 3: They should also explain the methodology used in the quantification (Table 2).

Response 3: It has been explained in the manuscript.

Changes in the manuscript (Lines192-194): A sentence describing the quantification method was added “The quantification of the phyto-compounds was performed by in-house developed and validated HPLC method (refer to section 3.5 for details).

Point 4: Although the authors used a positive control they almost ignore it during the discussion.

Response 4: Discussion on galantamine has been added.

Changes in the manuscript (Lines84-89): The following paragraph has been added “Galantamine, an isoquinoline alkaloid family, is a reversible and competitive inhibitor of AChE. It increases the level of ACh in the synaptic cleft, thus improving cholinergic transmission and improving neuron to neuron communications [22]. It has a dual action of mechanism; inhibiting AChE and allosterically modulating nicotinic acetylcholine receptor (nAChR) activity [23]. Galantamine shows 53-fold higher selectivity for human AChE than for butyrylcholinesterase (BChE) [24]”.

Point 5: They should explain why the extract “can be an alternative for the treatment” having higher v IC50values.

Response 5: AChE inhibitor ‘Galantamine’ is a pure compound; that is why its IC50 is lower compared to the extracts. But the extracts may contain thousands of different molecules in differing ratio, and the effective molecules in the extracts may be in lesser amount. That is why its IC50 is higher.

Changes in the manuscript (Lines421-424): Despite of having higher IC50 values compared to the tool compound, these mono-herbal extracts (may contain thousands of different phyto-compounds) were showing potency against AChE which should be further evaluated and could be used as cost effective alternatives for the treatment of AD.